# Enhanced Phonon Antibunching in a Circuit Quantum Acoustodynamical System Containing Two Surface Acoustic Wave Resonators

**DOI:** 10.3390/mi13040591

**Published:** 2022-04-09

**Authors:** Tai-Shuang Yin, Guang-Ri Jin, Aixi Chen

**Affiliations:** Department of Physics, Zhejiang Sci-Tech University, Hangzhou 310018, China; grjin@zstu.edu.cn

**Keywords:** surface acoustic wave, acoustic cavity quantum electrodynamics, phonon blockade

## Abstract

We propose a scheme to implement the phonon antibunching and phonon blockade in a circuit quantum acoustodynamical system containing two surface acoustic wave (SAW) resonators coupled to a superconducting qubit. In the cases of driving only one SAW resonator and two SAW resonators, we investigate the phonon statistics by numerically calculating the second-order correlation function. It is found that, when only one SAW cavity is resonantly driven, the phonon antibunching effect can be achieved even when the qubit–phonon coupling strength is smaller than the decay rates of acoustic cavities. This result physically originates from the quantum interference between super-Poissonian statistics and Poissonian statistics of phonons. In particular, when the two SAW resonators are simultaneously driven under the mechanical resonant condition, the phonon antibunching effect can be significantly enhanced, which ultimately allows for the generation of a phonon blockade. Moreover, the obtained phonon blockade can be optimized by regulating the intensity ratio of the two SAW driving fields. In addition, we also discuss in detail the effect of system parameters on the phonon statistics. Our work provides an alternative way for manipulating and controlling the nonclassical effects of SAW phonons. It may inspire the engineering of new SAW-based phonon devices and extend their applications in quantum information processing.

## 1. Introduction

Exploring the quantum behaviors of mechanical systems at a macroscopic scale is crucial not only for the fundamental research such as revealing the boundary between classical and quantum physics [1,2,3,4,5], but also for the practical applications in quantum information processing and quantum measurement [6,7,8]. On the one hand, with the significant developments of micro- and nanomechanical fabricating technologies, mechanical resonators with a wide range of frequencies could be experimentally cooled to their quantum ground state [9,10,11,12]. This achievement makes it possible for manipulating and controlling the mechanical resonators at the single phonon level. On the other hand, due to their ability to respond sensitively to weak electric, magnetic, and optical signals, mechanical resonators can be coupled to diverse quantum systems to form hybrid systems [13,14,15,16,17], e.g., the radiation pressure coupling between photons and phonons in cavity optomechanics [18,19,20,21], mechanical resonators coupled to superconducting qubits [22,23,24,25,26,27], and spins [28,29,30,31,32,33,34,35,36,37,38]. These coupled systems greatly facilitate the deep research of mechanical quantum effects, including the nonclassical states of phonons [2,3,4,38,39,40,41,42], mechanical squeezing, and entanglement [43,44,45,46,47,48] as well as phonon blockade [24,25,26,27,49,50,51,52,53,54].

However, most theoretical schemes and experimental implementation associated with mechanical quantum effects involve a bulk harmonic mode of mechanical resonators, including a drum mode of a membrane, or a flexural mode of a beam and a wire. Note that surface acoustic waves (SAWs) [55], as alternative quantum acoustic modes, have attracted considerable attention in past decades. SAWs are acoustic waves that propagate along the surface of a solid or an interface, which can be excited and probed by interdigitated transducers on piezoelectric substrates. They have long been used to engineer compact electric devices and applied into classical wireless communication [56]. In contrast to photons, SAW phonons have several striking features. First, their propagating speed is usually at a few km/s (five orders of magnitude slower than that of vacuum electromagnetic waves), which means shorter wavelengths for the same frequency. This feature makes acoustic phonons more advantageous in short distance quantum communication. For example, it allows for manipulating many-wavelength signals and in-flight control of SAWs [57]. Second, the typical frequency of SAWs is in the Gigahertz range, which closely matches the transition frequency of an artificial atom. More importantly, due to their high frequency, acoustic modes could be cooled to their quantum ground states by a conventional cryogenic refrigerator. This greatly sparks the extensive research of quantum acoustics [58] and the exploration of SAW-based applications in the quantum field [59,60,61,62]. In particular, recent experiments have demonstrated local probing of SAWs [63] and propagating phonons coupled to an artificial atom in the quantum regime [64]. Traveling acoustic waves have been used to observe electromagnetically induced transparency (EIT) of a mechanical field [65] and route single phonon [57]. In addition to being propagating phonons, SAWs can also be confined to acoustic cavities to form local acoustic modes.

High quality factor Fabry–Pérot SAW resonators (Q∼$10^{5}$) have been reported and experimentally realized [66,67,68,69]. It has been demonstrated that a SAW resonator can be strongly coupled to superconducting qubits through piezoelectric interactions [70,71,72,73,74], which ultimately allows for the realization of the quantum regime of a two-dimensional phonon cavity [73]. These developments open up the study of circuit quantum acoustodynamics (QAD) [71,75,76,77]. Analogous to circuit quantum electrodynamics (QED) [78,79,80], circuit QAD explores the interaction between SAW resonators (or acoustic waves) and artificial atoms, which paves a new direction for the exploration of phonon–matter interaction. By exploiting the strong coupling between SAW resonators and superconducting qubits, many achievements have been made, including the synthesis of mechanical Fock states and a Fock state superposition [74], a generation of phonon-mediated quantum state transfer and remote qubit entanglement [81], resolve phonon number states [82], and a potential quantum acousto-optic transducer [83]. In addition, SAW resonators can also be coupled to other quantum systems (e.g., quantum dots [84,85], the nitrogen-vacancy centers in diamond [86,87] and trapped ions [88]), which could be utilized as universal quantum transducers [88]. Moreover, the integration of SAW resonators with different physical systems, including optical cavity and microwave resonators, could form hybrid piezo-optomechanical systems, which has been harnessed to observe the enhanced anti-Stokes scattering [89], realize single-photon quantum regime of artificial radiation pressure on SAW phonons [90], and generate controllable optomechanically induced transparency [91].

Here, we propose a scheme for exploring the phonon antibunching effect and phonon blockade in a circuit quantum acoustodynamical system containing two SAW resonators. Specifically, we consider a system consisting of a superconducting transmon qubit coupled to two SAW resonators. It is found that, when only one SAW resonator is driven, an unexpectedly strong phonon antibunching effect with sub-Poissonian statistics is generated under the mechanical resonant driving even though the qubit–phonon coupling strength is smaller than the decay rates of acoustic cavities. This phenomenon can be physically explained from the combined effect of super-Poissonian statistics and Poissonian statistics, similar to that in quantum-dot-cavity QED for photons [92,93,94]. Comparing with the case of only driving one SAW resonator, when the two SAW resonators are both resonantly driven, the sub-Poissonian statistics of phonons can be significantly enhanced and a phonon blockade occurs. Moreover, by adjusting the driving strength ratio of the two SAW resonators, an optimal phonon antibunching and phonon blockade can be obtained. Our work provides a powerful platform, i.e., a circuit quantum acoustodynamical system to reveal phonon nonclassical effects, which may inspire the development of SAW-based single-phonon quantum devices and their applications in quantum information processing.

Note that phonon blockade, as a typical quantum effect, has been studied in many previous works, such as in quadratically coupled optomechanical systems [49,50,51,52] and qubit- and spin-mechanical resonator coupled systems [24,25,26,27,53,54]. These systems mainly focus on the localized mechanical modes of micro- and nanomechanical resonators. On the contrary, our work is based on the acoustic modes in surface acoustic waves. In terms of SAWs in acoustic cavities, a recent work in Ref. [95] proposed an effective N-type quantum system to realize vacuum-induced SAW phonon blockade. This work utilizes a hybrid quantum system consisting of a superconducting qubit coupled to a SAW resonator (or waveguide) and a microwave resonator. By employing the vacuum field of the microwave resonator, a giant self-Kerr phononic nonlinearity is induced to cause the SAW phonon blockade. Our work is quite different from this work. First, we consider the coupling between a superconducting qubit and two SAW resonators, which does not involve a microwave resonator. Second, in the case of mechanical resonant driving, the generation of the SAW phonon blockade in our proposal is due to the quantum interference between Poissonian statistics and super-Poissonian statistics of phonons, instead of a strong mechanical nonlinearity. In addition, the work in Ref. [96] studies the two-acoustic-cavity interaction, which is mediated by auxiliary transmons.

It shows that, by considering the transmon as a quantum two-level qubit and a three-level qutrit, one could observe the quantum state transfer between two SAW resonators and correlated phonon pairs. In stark contrast to these, our work is based on the SAW phonon–qubit coupling and concentrates on the nonclassical effects of SAW modes. Note that the works in Refs. [93,94] studied the sub-Poissonian light generation and the optimal photon antibunching in a quantum-dot-bimodal-cavity coupled system. The achieved nonclassical statistics of photons essentially originate from the quantum interference between a coherent state and a super-Poissonian state for the single-mode driving regime. For the case of two-mode driving, the photon antibunching effect of the additional driven cavity mode is also associated with the combined effect of the direct driving and the quantum-dot-cavity coupling. These works indeed inspire our research for the nonclassical effects of SAW phonons, which exploit similar physical mechanisms. However, first, it should be pointed out that our work is based on SAW phonons in acoustic cavities, which is very different from the previous works [93,94] focusing on photons. Second, utilizing the related physical mechanism in [93,94] to study the nonclassical effects of SAW phonons is currently not explored. Finally, when we discuss the influences of system parameters, it is found that the phonon antibunching effect is sensitive to the thermal phonon occupation numbers. However, SAW phonons have the feature of high frequencies, which provides an inherent advantage for suppressing the adverse effects of thermal noise, especially in practical experiments. Overall, our work provides an alternative approach for achieving the nonclassical effects of SAW phonons in a circuit quantum acoustodynamical system. It may deepen the understanding of phonon–matter interactions and inspire the engineering of SAW-based phonon quantum devices as well as their potential applications.

The system we consider consists of two SAW resonators coupled to a superconducting qubit. For a SAW resonator, two IDTs are introduced to excite and probe the SAW, respectively, and placed between two Bragg mirrors, displaying an acoustic Fabry–Pérot cavity. However, the proposed ’resonator’ asymmetric geometry (left–right) that introduces IDTs on both axes will determine significant wave reflection on the propagation surface, which should produce a lot of noise and oscillation band enlargement. Note that, first, the periodicity $\lambda_{IDT}$ of the IDTs determines the acoustic wavelength and the resonant frequency is given by $\omega =2\pi v_{0}/\lambda_{IDT}$, where $v_{0}$ is the SAW propagation speed. Many Fabry–Pérot standing modes are localized in the mirror stop band. However, resonances are only produced in the band where the mirrors strongly reflect. In addition, the SAW resonator can be designed to support a single SAW resonance in the mirror stop band [56,74]. Second, each of the device has a finite frequency bandwidth determined by its geometry. One can calculate frequency characteristics of different circuit elements and measure the acoustic response [73,80]. By performing the measurement of transmission amplitude or reflection coefficient versus the frequency of the acoustic resonator, one could observe resonances of different modes. In addition, the coupling strength between SAW modes and the qubit depends on the spatial overlap of a mode and qubit-IDT fingers [72]. The phonon–qubit coupling can be strongest by positioning the electrodes of the qubit-IDT at the antinode of standing acoustic wave in the resonator. When the qubit is tuned into resonance with the acoustic mode, one can investigate the anticrossing spectrum, which is used to measure the coupling strength. The energy splitting of avoided anticrossing does not appear in other modes. This is due to other modes corresponding to standing acoustic wave with nodes the qubit electrode locations.

This paper is organized as follows: In Section 2, we describe a circuit quantum acoustodynamical system consisting of a superconducting qubit coupled to two SAW resonators, and derive the corresponding Hamiltonian. In Section 3, we first analyze the generation of the phonon antibunching effect when only one SAW resonator is resonantly driven, and explain the origination of this phenomenon. Then, we present the case of simultaneously driving two SAW resonators. It is shown that the phonon antibunching effect can be significantly enhanced and the phonon blockade is generated. Furthermore, an optimal phonon blockade is obtained by adjusting the driving strength ratio between the driving fields for the two SAW resonators. We numerically investigate the phonon statistics characterized by the second-order correlation function, and discuss in detail the influence of system parameters on the phonon blockade effect. The work is summarized and conclusions are given in Section 4.

## 2. Theoretical Model

We consider a circuit quantum acoustodynamical system consisting of a superconducting qubit coupled to two SAW resonators. As shown in Figure 1, a SAW resonator is fabricated on piezoelectric substrates and defined by a pair of Bragg mirrors, forming an acoustic Fabry–Pérot cavity. Each acoustic cavity contains two interdigitated transducers (IDTs). These IDTs, situated inside the acoustic resonators, are formed by periodic arrays of identical electrodes, which can be used to excite and probe the SAW. A transmon qubit is located in the middle of the SAW resonator and coupled to the SAW mode via another IDT through the piezoelectric substrates. In addition, the transmon qubit lies in the intersection region of these two perpendicular SAW resonators so that the transmon qubit could be coupled to both of acoustic resonators [96]. The system Hamiltonian reads (ℏ=1)
(1)$$\begin{array}{ccc} H & = & \omega_{1}b_{1}^{†}b_{1}+\omega_{2}b_{2}^{†}b_{2}+\omega_{q}\sigma_{+}\sigma_{-}+g_{1}\left(b_{1}^{†}\sigma_{-}+b_{1}\sigma_{+}\right)\\ & & +g_{2}\left(b_{2}^{†}\sigma_{-}+b_{2}\sigma_{+}\right), \end{array}$$
where $b_{i}$ ($b_{i}^{†}$) is the annihilation (creation) operator of the acoustic mode in the i(i=1,2) SAW resonator. The first two terms in Equation (Equation 1) represent the free Hamiltonian of the two SAW resonators with resonant frequency $\omega_{i}$. The third term is the Hamiltonian of the superconducting qubit with frequency $\omega_{q}$. The last two terms describe the piezoelectric coupling between the SAW modes and the qubit with the coupling strength $g_{i}$. For simplicity, we assume that the frequencies of the two acoustic cavities and the qubit are the same, i.e., $\omega_{1}=\omega_{2}=\omega_{q}$, and the SAW-qubit coupling strengths are the same, i.e., $g_{1}=g_{2}=g$.

Including the dissipation caused by the system–bath coupling, the dissipative dynamics of the hybrid SAW-qubit system are described by the master equation
(2)$$\begin{array}{ccc} \dot{\rho } & = & -i\left[H_{\mathrm{tot}},\rho \right]+\kappa \left(n_{\mathrm{th}}+1\right)D\left[b_{1}\right]\rho +\kappa n_{\mathrm{th}}D\left[b_{1}^{†}\right]\rho \\ & & +\kappa \left(n_{\mathrm{th}}+1\right)D\left[b_{2}\right]\rho +\kappa n_{\mathrm{th}}D\left[b_{2}^{†}\right]\rho +\gamma D\left[\sigma_{-}\right]\rho , \end{array}$$
where $D\left[o\right]\rho =o\rho o^{†}-\frac{1}{2}\left(o^{†}o\rho +\rho o^{†}o\right)$ is the standard Lindblad superoperator, κ and γ are the decay rates of the two acoustic cavities and the qubit, respectively, and $n_{\mathrm{th}}$ is the thermal phonon occupation number with $n_{\mathrm{th}}={\left[\exp\left(\hbar \omega_{i}/k_{B}T\right)-1\right]}^{-1}$, where $k_{B}$ is Boltzmann constant and *T* is the temperature of the thermal reservoir. Note that, for simplicity, in Equation (Equation 2), we have assumed that the decay rates of two acoustic resonators are the same (i.e., κ) and the thermal phonon occupation numbers are the same (i.e., $n_{\mathrm{th}}$). Based on Equation (Equation 2), the statistical properties of the phonons are investigated by numerically calculating the correlation function. The presence of IDTs and measurement port leads to external losses, which contributes to the external quality factor of SAW resonators. By performing measurements, the expression of the complex reflection coefficient is both related with the external quality factor and the internal quality factor (intrinsic to the resonators) [66,67,68]. Note that high quality factor ($Q_{i}$) Fabry–Pérot SAW resonators have been reported and experimentally realized [66,67,68,69]. In the current manuscript, we mainly focus on how the phonon nonclassical effect is achieved and the influence of system dissipations. Thus, we did not study the effect of different dissipation components separately, but considered the overall dissipation.

To exhibit the quantum behavior of the phonons, we investigate the statistical properties of the phonons, which can be characterized by the equal-time second-order correlation function in the steady state
(3)$$g_{b_{i}}^{\left(2\right)}\left(0\right)={\mathrm{Lim}}_{t\to \infty }\frac{\langle b_{i}^{†}b_{i}^{†}b_{i}b_{i}\rangle \left(t\right)}{{\langle b_{i}^{†}b_{i}\rangle }^{2}\left(t\right)}.$$

When $g_{b_{i}}^{\left(2\right)}\left(0\right)<1$ ($g_{b_{i}}^{\left(2\right)}\left(0\right)\to 0$), it indicates that the phonon antibunching effect (phonon blockade) with the sub-Poissonian statistics occurs. In the following, we will present how the phonon antibunching and phonon blockade can be achieved by driving SAW resonators.

## 3. Results

### 3.1. The Phonon Antibunching When Only One SAW Resonator Is Driven

To observe the phonon statistical properties, we first consider a weak probe field (with frequency $\omega_{l}$ and strength $\varepsilon_{b_{i}}$) applied into one of two SAW resonators (e.g., the i=1 resonator) and the corresponding Hamiltonian is $H_{p_{1}}=\varepsilon_{b_{1}}\left(b_{1}^{†}e^{-i\omega_{l}t}+b_{1}e^{i\omega_{l}t}\right)$. In a frame rotating with frequency $\omega_{l}$ for the acoustic modes and the qubit system, the total Hamiltonian can be written as
(4)$$\begin{array}{ccc} H_{\mathrm{tot}} & = & \Delta \left(b_{1}^{†}b_{1}+b_{2}^{†}b_{2}+\sigma_{+}\sigma_{-}\right)+g_{1}\left(b_{1}^{†}\sigma_{-}+b_{1}\sigma_{+}\right)\\ & & +g_{2}\left(b_{2}^{†}\sigma_{-}+b_{2}\sigma_{+}\right)+\varepsilon_{b_{1}}\left(b_{1}^{†}+b_{1}\right), \end{array}$$
where the detuning is $\Delta =\omega_{i}-\omega_{l}=\omega_{q}-\omega_{l}$. As mentioned above, Refs. [93,94] studied the photon nonclassical statistics in a quantum-dot-bimodal-cavity system. The physical mechanism we exploits in the SAW phonon–qubit coupled system is similar to that in Refs. [93,94] for photons. Thus, it should be made clear that the relevant analysis and discussions for achieving the SAW phonon antibunching effect are not a totally novel result.

In Figure 2, we calculate the steady-state equal-time second-order correlation function $g_{b_{1}}^{\left(2\right)}\left(0\right)$ of the SAW mode $b_{1}$ versus the driving detuning Δ/κ by numerically solving the master equation in Equation (Equation 2) with the total Hamiltonian $H_{\mathrm{tot}}$. It is shown that, when the coupling strength is strong, e.g., g/κ=2, the phonon antibunching effect appears only around the detuning $\Delta \approx \pm \sqrt{2}g$, which corresponds to the single-phonon resonant condition for the polariton of the coupled system. However, when the value of g/κ gradually decreases, a significant phonon antibunching effect is generated at the resonant driving Δ=0. Specifically, compared to the case of g/κ=2, when g/κ=1, one still observes the phonon antibunching effect at $\Delta \approx \pm \sqrt{2}g$ even though this effect becomes weaker. Unexpectedly, sub-Poissonian statistics with $g_{b_{1}}^{\left(2\right)}\left(0\right)<1$ are observed around the detuning Δ=0. Note that there are two dips with a minimum of $g_{b_{1}}^{\left(2\right)}\left(0\right)$, which is associated with the two-phonon blockade. With g/κ decreasing to even the weak coupling regime (e.g., g/κ=0.5 and g/κ=0.3), the sub-Poissonian characteristics is much stronger at the detuning Δ=0. It is this regime for the two acoustic resonators-qubit system that we will focus on in this paper.

To fully illustrate the dependence of the sub-Poissonian phonon statistics on the coupling strength *g*, the acoustic decay rate κ and the driving detuning Δ, in Figure 3, we plot $g_{b_{1}}^{\left(2\right)}\left(0\right)$ versus the ratio g/κ and the driving detuning Δ/κ. In this figure, the white solid line, the white dashed line, and the black dotted line correspond to the cases of $g_{b_{1}}^{\left(2\right)}\left(0\right)=0.1,\;0.4,\;0.482$, respectively. It can be seen that, when *g* is comparable to κ (i.e., g/κ∼1) or even in the weak-coupling regime (i.e., g/κ<1), a notable effect of phonon antibunching occurs around the detuning Δ=0 with the optimal coupling g/κ≈0.3. This parameter regime is different from that for the conventional quantum effects of phonons and cannot be explained by the the anharmonic nature of the coupled system.

To understand the origin of this abnormal quantum phenomenon, we introduce a mode basis $\alpha =\left(b_{1}+b_{2}\right)/\sqrt{2},\;\beta =\left(b_{1}-b_{2}\right)/\sqrt{2}$. Then, in terms of α and β, the Hamiltonian in Equation (Equation 4) can be written as $H_{\mathrm{tot}}=H_{1}+H_{2}$, where
(5)$$\begin{array}{ccc} H_{1}\; & = & \;\Delta \left(\alpha^{†}\alpha +\sigma_{+}\sigma_{-}\right)\;+\;\sqrt{2}g\left(\alpha^{†}\sigma_{-}+\alpha \sigma_{+}\right)\;+\;\frac{\varepsilon_{b_{1}}}{\sqrt{2}}\left(\alpha^{†}+\alpha \right),\\ H_{2}\; & = & \;\Delta \beta^{†}\beta \;+\;\frac{\varepsilon_{b_{1}}}{\sqrt{2}}\left(\beta^{†}+\beta \right). \end{array}$$

The Hamiltonian $H_{1}$ describes a single acoustic cavity–qubit coupled system. The Hamiltonian $H_{2}$ represents a driven empty cavity without the qubit coupling. Both acoustic cavities are resonantly driven with the same driving strength $\varepsilon_{b_{1}}/\sqrt{2}$. In the transformed basis, one can have $b_{1}=\left(\alpha +\beta \right)/\sqrt{2}$. Therefore, the statistical character of mode $b_{1}$ is equivalent to the combination of mode α and β. As shown in Figure 4b, we plot the steady-state equal-time second-order correlation function $g^{\left(2\right)}\left(0\right)$ of modes α, β and $b_{1}$ versus the driving detuning Δ/κ. On the one hand, for the coupled acoustic cavity (α), the correlation function is $g^{\left(2\right)}\left(0\right)\gg 1$ at the resonant driving Δ=0, which clearly presents the super-Poissonian phonon statistics. This is due to phonon-induced tunneling, which is similar to that in quantum-dot-cavity QED for photons [92]. On the other hand, for the empty acoustic cavity (β), Poissonian statistics are presented with $g^{\left(2\right)}\left(0\right)=1$. Combining these two aspects, the output of mode $b_{1}$ shows sub-Poissonian statistics with $g^{\left(2\right)}\left(0\right)<1$ at the detuning Δ=0. Actually, this unexpected antibunching effect has been studied for photons in a quantum-dot-bimodal-cavity system and explained from the perspective of super-Poissonian light and coherent light [93,94]. In addition, in Figure 4a, we plot the second-order correlation function $g_{b_{1}}^{\left(2\right)}\left(0\right)$ versus the ratio g/κ for different driving strengths $\varepsilon_{b_{1}}/\kappa$ in the case of mechanical resonance Δ=0. It can be seen that, at a certain optimal coupling $g_{\mathrm{opt}}/\kappa <1$, $g_{b_{1}}^{\left(2\right)}\left(0\right)$ could have a minimum value, which indicates a strong phonon antibunching effect. This is consistent with the results obtained in Figure 3. Furthermore, comparing the three cases of $\varepsilon_{b_{1}}/\kappa =0.01$, 0.05, and 0.1, one could find that, when the driving strength $\varepsilon_{b_{1}}/\kappa$ decreases, the phonon antibunching effect is enhanced and the corresponding optimal coupling is slightly different.

In the discussion above, we do not consider the effect of the thermal phonon occupation number $n_{\mathrm{th}}$ (i.e., $n_{\mathrm{th}}=0$). To fully illustrate how the phonon statistics are influenced by the thermal phonon number, in Figure 5, we plot the steady-state equal-time second-order correlation function $g_{b_{1}}^{\left(2\right)}\left(0\right)$ of the SAW mode $b_{1}$ versus the coupling strength g/κ with different thermal phonon occupation numbers $n_{\mathrm{th}}$. It is shown that, when the thermal phonon occupation number $n_{\mathrm{th}}$ increases from $10^{-5}$ to $2\times 10^{-4}$, the minimum value of $g_{b_{1}}^{\left(2\right)}\left(0\right)$ is increased, and the curve is gradually moving up. It means that the increase of the thermal phonon number will destroy the generation of the phonon nonclassical effect. To fully demonstrate this, we present $g_{b_{1}}^{\left(2\right)}\left(0\right)$ as a function of the thermal phonon occupation number $n_{\mathrm{th}}$ (see the inset of Figure 5). It can be seen that the thermal noise has a significant effect on the phonon antibunching characteristics. When $n_{\mathrm{th}}=10^{-4}$, there still exists a strong phonon antibunching with the optimal $g_{b_{1}}^{\left(2\right)}\left(0\right)\approx 0.51$. When $n_{\mathrm{th}}$ reaches $6\times 10^{-4}$, the phonon antibunching effect will disappear. Therefore, it is necessary to suppress the thermal noise to observe the phonon nonclassical effect of the SAW mode.

### 3.2. Enhanced Phonon Antibunching When the Two SAW Resonators Are Both Driven

In this section, we will discuss the situation where the two SAW resonators are both driven. That is, in addition to driving the i=1 resonator, we consider another probe field (with frequency $\omega_{l}$ and strength $\varepsilon_{b_{2}}$) applied into the i=2 resonator and the corresponding Hamiltonian is $H_{p_{2}}=\varepsilon_{b_{2}}\left(b_{2}^{†}e^{-i\omega_{l}t}+b_{2}e^{i\omega_{l}t}\right)$. In a frame rotating with frequency $\omega_{l}$, the total Hamiltonian of the coupled system becomes
(6)$$H_{\mathrm{tot}}^{\prime }=H_{\mathrm{tot}}+\varepsilon_{b_{2}}\left(b_{2}^{†}+b_{2}\right),$$
and we calculate the steady-state equal-time second-order correlation function $g_{b_{i}}^{\left(2\right)}\left(0\right)\left(i=1,2\right)$ by numerically solving the master equation in Equation (Equation 2), but with the Hamiltonian $H_{\mathrm{tot}}^{\prime }$.

As shown in Figure 6, we plot the correlation function $g_{b_{i}}^{\left(2\right)}\left(0\right)$ and phonon number $N_{b_{i}}=\langle b_{i}^{†}b_{i}\rangle$ of the SAW modes $b_{1}$, $b_{2}$ versus the driving detuning Δ/κ for different driving strengths $\varepsilon_{b_{2}}$. It can be seen from Figure 6a,b that, when the i=2 resonator is not driven (i.e., $\varepsilon_{b_{2}}=0$), the phonon antibunching effect for mode $b_{1}$ can be generated around the detuning Δ=0, as discussed above. For the mode $b_{2}$, it also shows antibunching characteristics around Δ=0. However, this characteristic can only originate from the interaction with the qubit. Therefore, the sub-Poissonian character of mode $b_{2}$ is similar to that generated by resonant excitation of a qubit or an effective two-level system [93]. Differently, when the driving strength is applied, e.g., $\varepsilon_{b_{2}}=0.5\varepsilon_{b_{1}}$, the values of the correlation function for modes $b_{1}$ and $b_{2}$ become much smaller at Δ=0 with $g_{b_{i}}^{\left(2\right)}\left(0\right)\ll 1$. It indicates that the phonon antibunching effect can be significantly enhanced by simultaneously driving two acoustic resonators instead of one resonator. However, when the driving strength further increases, e.g., $\varepsilon_{b_{2}}=0.6\varepsilon_{b_{1}}$, the value of $g_{b_{i}}^{\left(2\right)}\left(0\right)$ increases instead at Δ=0. That is, the antibunching effect of phonons is weakened. These results suggest that, to obtain an enhanced phonon antibunching effect, only driving the two SAW resonators is not enough. Importantly, it needs to control the two driving strengths $\varepsilon_{b_{2}}$ and $\varepsilon_{b_{1}}$ so that the enhanced result is optimal.

To further study how the statistical characteristics of phonons are influenced by the driving strengths, we define the ratio of two driving strengths as $R=\varepsilon_{b_{2}}/\varepsilon_{b_{1}}$. In Figure 7, the steady-state equal-time second-order correlation function $g^{\left(2\right)}\left(0\right)$ and the phonon number *N* of the two SAW modes $b_{1}$ and $b_{2}$ are plotted as a function of the ratio *R* in the mechanical resonant case Δ=0. It is found that, with the ratio *R* increasing, the values of $g^{\left(2\right)}\left(0\right)$ first decrease and then increase. Thus, the curves of the correlation function have a dip with a minimum of $g^{\left(2\right)}\left(0\right)$ for both modes, as shown in Figure 7a,c,e. This clearly illustrates that, by adjusting the ratio of the driving strengths, one can obtain an optimal phonon antibunching effect with $g^{\left(2\right)}\left(0\right)\ll 1$. Moreover, the value of $g_{b_{i}}^{\left(2\right)}\left(0\right)$ at the optimal driving ratio $R_{\mathrm{opt}}$ is much smaller than that in the case of R=0. Note that, for modes $b_{1}$ and $b_{2}$, the corresponding optimal driving ratio $R_{\mathrm{opt}}$ is only slightly different. Furthermore, the minimum $g^{\left(2\right)}\left(0\right)$ of mode $b_{2}$ is more than one order of magnitude smaller than that of mode $b_{1}$. For example, in Figure 7e, the minimum value of the correlation function for mode $b_{2}$ is $g_{b_{2}}^{\left(2\right)}\left(0\right)=0.0005$, which is almost approaching 0. This indicates that the sub-Poissonian statistical statistics of phonons are significantly enhanced and the phonon blockade effect is generated. In addition, Figure 7a,c,e also show the second-order correlation function $g^{\left(2\right)}\left(0\right)$ for different coupling strengths g/κ. It can be seen that, for an appropriate coupling strength, there always exists an optimal driving ratio $R_{\mathrm{opt}}$, where a strong phonon antibunching and even phonon blockade can be obtained. In addition, one could find that, when the coupling strength changes, the corresponding optimal value $R_{\mathrm{opt}}$ has a little shift.

In order to better explain the physics behind the figures in the case of simultaneously driving two acoustic resonators, we will start with the introduced mode basis α,β (see the previous section). When the driving field $\varepsilon_{b_{2}}$ is applied into the i=2 SAW resonator, in terms of α and β, the total Hamiltonian in Equation (Equation 6) can be transformed into $H_{\mathrm{tot}}^{\prime }=H_{1}^{\prime }+H_{2}^{\prime }$, where $H_{1}^{\prime }=\Delta \left(\alpha^{†}\alpha +\sigma_{+}\sigma_{-}\right)+\sqrt{2}g\left(\alpha^{†}\sigma_{-}+\alpha \sigma_{+}\right)+E_{\alpha }\left(\alpha^{†}+\alpha \right)$, and $H_{2}^{\prime }=\Delta \beta^{†}\beta +E_{\beta }\left(\beta^{†}+\beta \right)$. Here, $E_{\alpha }=\left(1+R\right)\varepsilon_{b_{1}}/\sqrt{2}$ and $E_{\beta }=\left(1-R\right)\varepsilon_{b_{1}}/\sqrt{2}$ denote the effective driving strengths for the phonon–qubit coupled cavity (α) and the empty acoustic cavity (β), respectively, which are different from that in the case of driving only acoustic resonator. When R=0, the driving strengths of both acoustic cavities are the same (see Equation (Equation 5)). In the presence of the driving field $\varepsilon_{b_{2}}$, the driving strengths of both acoustic cavities are different. As mentioned above, the statistical property of mode $b_{1}$ can be regarded as the combination of super-Poissonian phonon statistics for the mode α and Poissonian phonon statistics for the mode β. Even though the ratio *R* has a minor effect on the statistical properties of modes α and β, it does provide an opportunity to adjust both statistical components. This ultimately allows for the generation of an optimal phonon antibunching effect, as shown in Figure 6a,c and Figure 7a,c,e. For mode $b_{2}$, the optimal strategy is achieved by regulating the combined ratio between direct driving of mode $b_{2}$ and phonon exchange via the qubit–phonon coupling, and only the former component is associated with the ratio *R*. However, the contribution of modes α and β to the statistical feature of mode $b_{1}$ is simultaneously regulated via the driving ratio *R*. This difference leads to the different dependence of the second-order correlation function $g_{b_{i}}^{\left(2\right)}\left(0\right)$ of the SAW modes $b_{1}$ and $b_{2}$ on the system parameters. For example, in Figure 7a,c,e, the values of $g_{b_{i}}^{\left(2\right)}\left(0\right)$ and the corresponding optimal ratio $R_{\mathrm{opt}}$ for the modes $b_{1}$ and $b_{2}$ are slightly different. The following text will also discuss this phenomenon in detail. In addition, in Figure 6b,d, one can find that, when the ratio *R* of two driving strengths increases, the phonon number *N* of the two SAW modes $b_{1}$ and $b_{2}$ decreases in the mechanical resonant case Δ=0. This result can be investigated more clearly in Figure 7b,d,f. From the perspective of modes α and β, when the driving ratio *R* increases, the phonon number of mode β deceases. This can be easily obtained by the driven strength $E_{\beta }$ for the empty acoustic cavity (β). On the other hand, when the driving ratio *R* changes, the phonon number of mode α almost stays at 0. Therefore, with *R* increasing, the phonon numbers of both modes $b_{1}$ and $b_{2}$ decrease.

As the driving strength ratio $\varepsilon_{b_{2}}/\varepsilon_{b_{1}}$ is crucial to the generation of nonclassical properties of phonons, the influence of the driving strength $\varepsilon_{b_{1}}$ also needs to be considered. As shown in Figure 8a,b, we plot the equal-time second-order correlation function $g_{b_{i}}^{\left(2\right)}\left(0\right)$ of modes $b_{1}$ and $b_{2}$ versus the driving ratio *R* for different driving strengths $\varepsilon_{b_{1}}/\kappa$ in the mechanical resonant case Δ=0. It can be seen that, when the driving strength $\varepsilon_{b_{1}}/\kappa$ increases, the minimum value of $g_{b_{i}}^{\left(2\right)}\left(0\right)$ increases and the curves become wider and shallower. It reveals that, for obtaining a strong phonon antibunching and phonon blockade effect, a relatively weak driving strength for the mode $b_{1}$ should be favorable. However, one could find that the change of the driving intensity $\varepsilon_{b_{1}}/\kappa$ has a little influence on the optimal driving ratio $R_{\mathrm{opt}}$. In addition, by comparing Figure 8a,b, it can be seen that the effect of the driving $\varepsilon_{b_{1}}$ on the optimal $R_{\mathrm{opt}}$ of mode $b_{2}$ is relatively weaker. This is because the driving $\varepsilon_{b_{1}}$ for mode $b_{1}$ does not directly driving mode $b_{2}$. In order to further clarify the dependence of the phonon nonclassical characters on the inherent parameter condition, in Figure 8c,d, we illustrate the correlation function $g_{b_{i}}^{\left(2\right)}\left(0\right)$ as a function of the driving ratio *R* for different qubit decay rates γ/κ. Note that, when the qubit decay rate ranges from γ=0.05κ to γ=2κ, the optimal driving ratio has a significant shift, which is clearer in Figure 8c than that in Figure 8d. However, the corresponding minimum value of $g_{b_{i}}^{\left(2\right)}\left(0\right)$ only changes a little, especially for the mode $b_{1}$. On the other hand, one can find that, even though the qubit decay rate is larger than the decay of acoustic resonators (e.g.,γ=2κ), there still exists a strong phonon antibunching effect with $g^{\left(2\right)}\left(0\right)<1$ and even phonon blockade with $g^{\left(2\right)}\left(0\right)\to 0$. This illustrates that the qubit decay rate γ has a weak influence on the nonclassical characteristics of phonons. This is useful for relaxing the parameter condition of observing the phonon quantum effects, and thus provides an advantage for the experimental feasibility.

Next, we will discuss the influence of the thermal phonon occupation number $n_{\mathrm{th}}$ on the phonon nonclassical effect in the case of driving two SAW resonators. As presented in Figure 9a,b, we show the equal-time second-order correlation function $g_{b_{i}}^{\left(2\right)}\left(0\right)$ of modes $b_{1}$ and $b_{2}$ versus the driving ratio *R* for different thermal phonon occupation numbers $n_{\mathrm{th}}$. It can be seen that the presence of thermal phonon occupation number will significantly weaken the phonon blockade effect. Moreover, the insets in Figure 9a,b present $g_{b_{i}}^{\left(2\right)}\left(0\right)$ of modes $b_{1}$ and $b_{2}$ versus the thermal phonon occupation number $n_{\mathrm{th}}$, respectively. Clearly, the phonon quantum effect is sensitive to the thermal noise of the mechanical environment, which is similar to the results discussed in Figure 5. Thus, to better obtain the desired phonon nonclassical effect, it requires cooling the SAW resonators to low temperature. Fortunately, due to the high frequency of SAW modes, the acoustic modes can be cooled to their quantum ground state with the dilution refrigerator. For instance, at the base temperature T=16.5mK, when the frequency of a SAW mode is $\omega_{i}/\left(2\pi \right)=4\;\mathrm{GHz}$, the thermal phonon number is $n_{\mathrm{th}}\approx 10^{-6}$. Therefore, the high frequency of SAW resonators provides an advantage and a strong guarantee for observing the quantum effects of SAW phonons.

Finally, let us discuss the applicability of the proposed system. In our proposal, we exploit the piezoelectric coupling between two SAW resonators and a superconducting transmon qubit to achieve the desired phonon antibunching effect and phonon blockade. In principle, the same theoretical results also apply to other acoustic cavity-qubit systems, such as those contain superconducting charge, flux, or phase qubits. Note that, in current experiments, the type of superconducting qubits that has been successfully coupled to a SAW cavity is the transmon [70,71,72,73,74]. Other qubits, e.g., quantum dots and the nitrogen-vacancy centers in diamond, are generally coupled to a propagating SAW [84,85,86,87]. Nevertheless, we believe that, with the further developments of experimental technologies, more fields will be explored in quantum acoustics and breakthroughs will be made in the future.

## 4. Conclusions

In conclusion, we propose a scheme to achieve the phonon antibunching effect and phonon blockade of SAW modes in a circuit quantum acoustodynamical system, which consists of two SAW resonators coupled to a superconducting qubit. In the case of only driving one SAW resonator, a strong phonon antibunching is generated under the mechanical resonant condition even when the qubit–phonon coupling strength is smaller than the decay rates of acoustic cavities. This result originates from the quantum interference between super-Poissonian phonon statistics and Poissonian phonon statistics. In particular, when the two SAW resonators are simultaneously driven, the phonon antibunching can be significantly enhanced so that the phonon blockade appears. In addition, by controlling the driving strength ratio between two mechanical driving fields, one could obtain an optimal phonon blockade. This controllability provides an advantage for the manipulation of quantum effects of SAW phonons. We characterize the phonon statistics by numerically calculating the second-order correlation function. Furthermore, the effect of system parameters on the phonon statistics is also discussed in detail. Our work provides a promising approach to yield the typical quantum effects (i.e., phonon blockade) of SAW phonons. It could deepen the study for the interaction of phonons and matter in circuit QAD and extend the engineering of single phonon quantum devices based on SAWs as well as their applications in quantum information processing.

Finally, let us discuss the range of parameters in terms of state-of-the-art technology. Note that, in our proposal, to achieve the phonon antibunching effect and phonon blockade of SAW modes, the phonon–qubit coupling strength *g* is required to be comparable with the SAW decay rate κ and larger than the qubit decay rate γ. In terms of the experimental feasibility, the strong coupling regime (g≫κ) for the coupled SAW-qubit system has been demonstrated [72,81]. For example, the work in Ref. [72] shows the coupling strength g∼6.5 MHz, which exceeds the cavity loss rate (∼200 kHz) and qubit linewidth (∼1.1 MHz). The work in Ref. [81] reports a strong coupling system with g/2π∼2.5 MHz, κ/2π∼133 kHz and γ/2π∼76 kHz.

## Figures and Tables

**Figure 1 micromachines-13-00591-f001:**
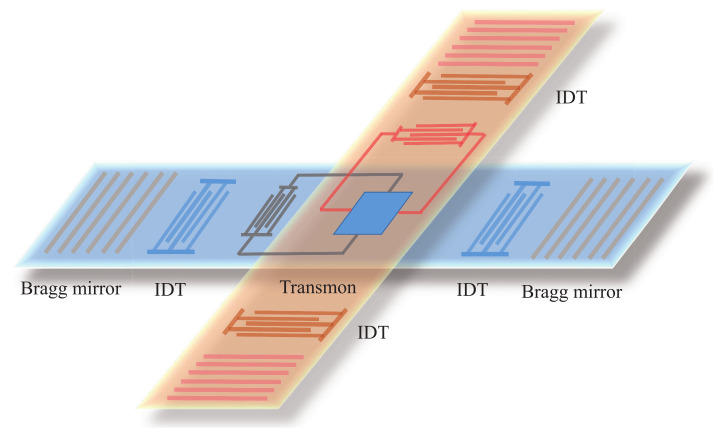
Schematic illustration of the proposed circuit quantum acoustodynamical system consisting of a superconducting transmon qubit coupled to two SAW resonators. Each SAW resonator is defined by a pair of Bragg mirrors and contains two identical interdigitated transducers (IDTs). These IDTs, situated inside the acoustic resonators, are formed by periodic arrays of identical electrodes, which can be used to excite and probe the SAW. A transmon qubit is located in the middle of a SAW resonator and coupled to the SAW mode via an IDT through the piezoelectric substrates. In addition, the transmon qubit lies in the intersection region of these two perpendicular SAW resonators so that the transmon qubit could be coupled to both of the acoustic resonators.

**Figure 2 micromachines-13-00591-f002:**
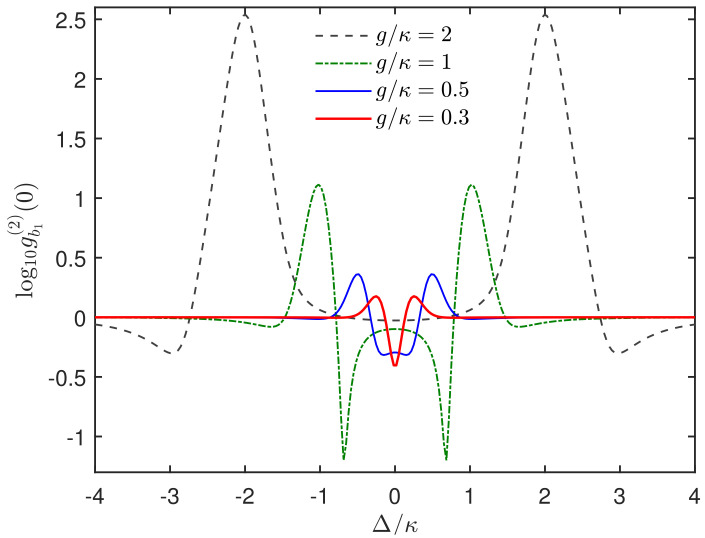
Steady-state equal-time second-order correlation function $g_{b_{1}}^{\left(2\right)}\left(0\right)$ of the SAW mode $b_{1}$ versus the mechanical driving detuning Δ/κ for different coupling strengths g/κ. The system parameters we take are γ=0.05κ, $\varepsilon_{b_{1}}=0.05\kappa$, $n_{\mathrm{th}}=0$.

**Figure 3 micromachines-13-00591-f003:**
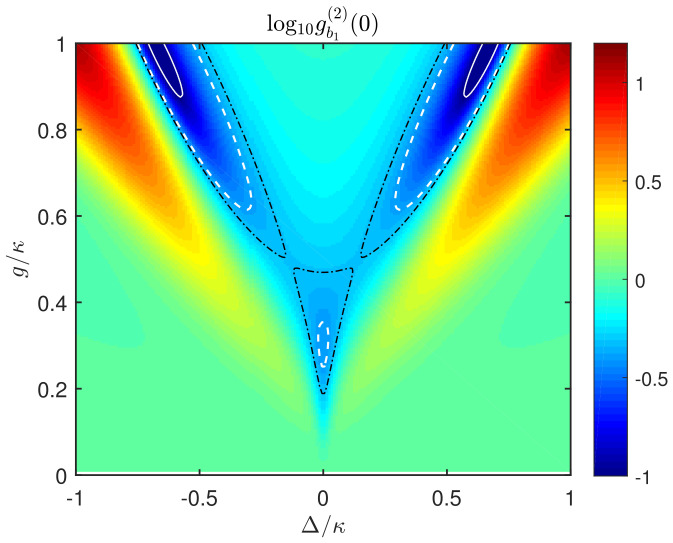
Steady-state equal-time second-order correlation function $g_{b_{1}}^{\left(2\right)}\left(0\right)$ of the SAW mode $b_{1}$ versus the mechanical driving detuning Δ/κ and the coupling strength g/κ. The system parameters are the same as in Figure 2. Here, the white solid line, the white dashed line, and the black dotted line correspond to the cases of $g_{b_{1}}^{\left(2\right)}\left(0\right)=0.1$, $g_{b_{1}}^{\left(2\right)}\left(0\right)=0.4$, and $g_{b_{1}}^{\left(2\right)}\left(0\right)=0.482$, respectively.

**Figure 4 micromachines-13-00591-f004:**
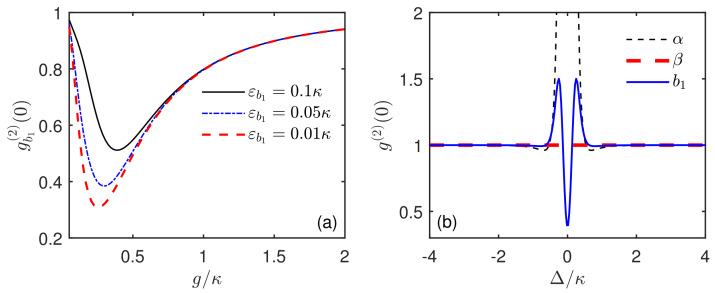
(**a**) Steady-state equal-time second-order correlation function $g_{b_{1}}^{\left(2\right)}\left(0\right)$ of the SAW mode $b_{1}$ versus the coupling strength g/κ with different driving strengths $\varepsilon_{b_{1}}/\kappa$ in the mechanical resonant case Δ=0; (**b**) steady-state equal-time second-order correlation function $g^{\left(2\right)}\left(0\right)$ of modes α, β, and $b_{1}$ versus the mechanical driving detuning Δ/κ. The system parameters are the same as in Figure 2 except for (**b**) g=0.3κ.

**Figure 5 micromachines-13-00591-f005:**
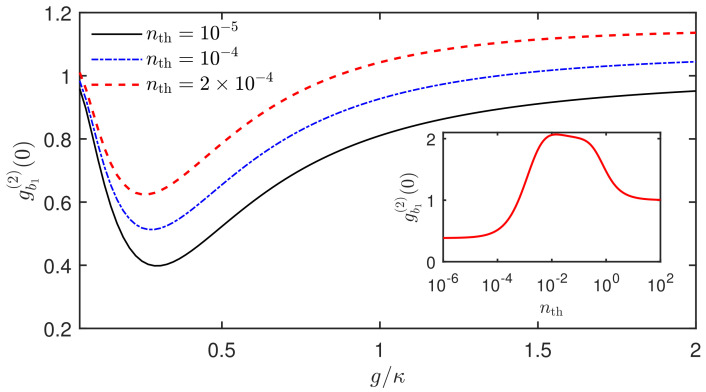
Steady-state equal-time second-order correlation function $g_{b_{1}}^{\left(2\right)}\left(0\right)$ of the SAW mode $b_{1}$ versus the coupling strength g/κ with different thermal phonon occupation numbers $n_{\mathrm{th}}$. The inset shows $g_{b_{1}}^{\left(2\right)}\left(0\right)$ versus the thermal phonon occupation number $n_{\mathrm{th}}$. The figure is plotted in the mechanical resonant case Δ=0. The system parameters are the same as in Figure 2 except for g=0.3κ for the inset.

**Figure 6 micromachines-13-00591-f006:**
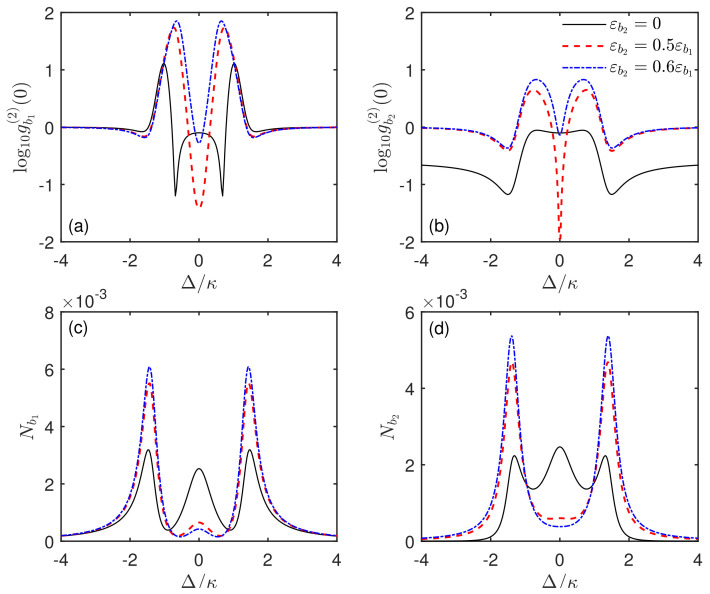
(**a**,**b**) Steady-state equal-time second-order correlation function $g_{b_{i}}^{\left(2\right)}\left(0\right)$ and (**c**,**d**) the phonon number $N_{b_{i}}$ of the SAW modes $b_{1}$ and $b_{2}$ versus the driving detuning Δ/κ with different driving strengths $\varepsilon_{b_{2}}$. Here, the black solid line, the red dashed line, and the blue dotted line correspond to the cases of $\varepsilon_{b_{2}}=0$, $\varepsilon_{b_{2}}=0.5\varepsilon_{b_{1}}$, and $\varepsilon_{b_{2}}=0.6\varepsilon_{b_{1}}$, respectively. The system parameters are the same as in Figure 2, except for g=κ.

**Figure 7 micromachines-13-00591-f007:**
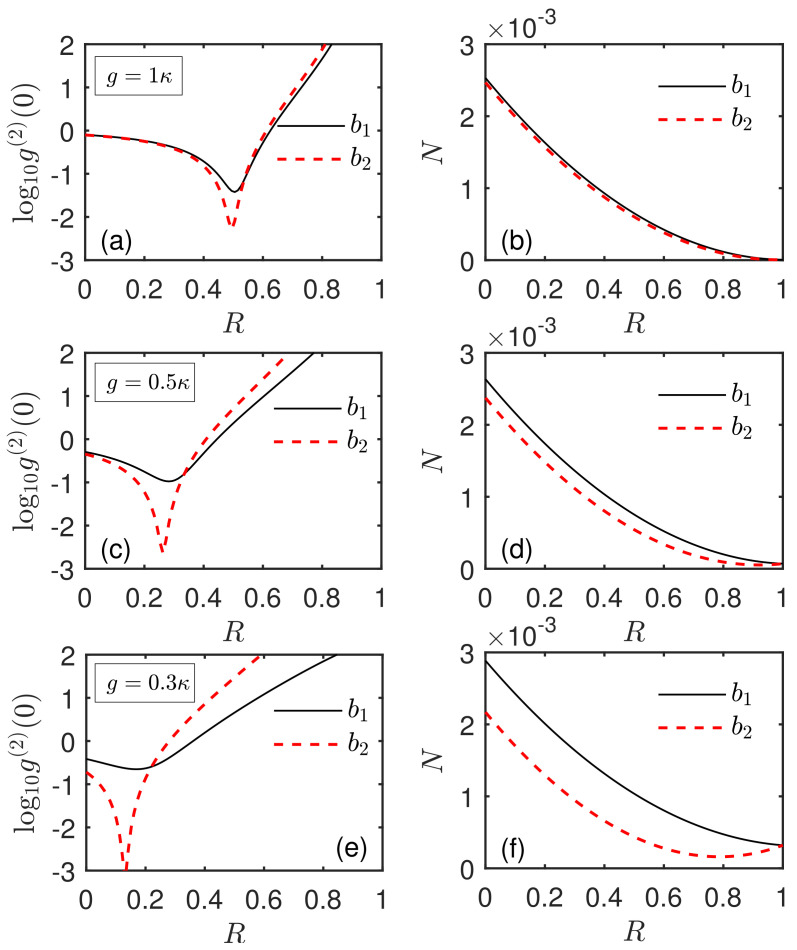
(**a**,**c**,**e**) Steady-state equal-time second-order correlation function $g^{\left(2\right)}\left(0\right)$, and (**b**,**d**,**f**) the phonon number *N* of the two SAW modes $b_{1}$ and $b_{2}$ versus the ratio *R* of two driving strengths in the mechanical resonant case Δ=0. Here, the black solid line and the red dashed line correspond to the cases of modes $b_{1}$ and $b_{2}$, respectively. The system parameters are the same as in Figure 2 except for (**a**,**b**) g=κ, (**c**,**d**) g=0.5κ, and (**e**,**f**) g=0.3κ.

**Figure 8 micromachines-13-00591-f008:**
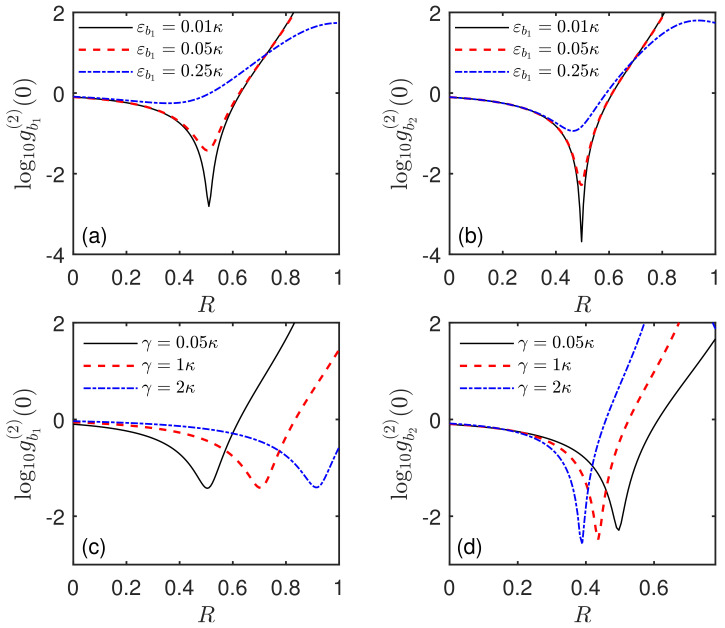
Steady-state equal-time second-order correlation function $g_{b_{i}}^{\left(2\right)}\left(0\right)$ of the SAW mode $b_{1}$ and $b_{2}$ versus the ratio *R* of two driving strengths (**a**,**b**) for different driving strengths $\varepsilon_{b_{1}}/\kappa$ and (**c**,**d**) for different qubit decay rates γ/κ. These results are all obtained in the mechanical resonant case Δ=0. The system parameters are g=κ, $n_{\mathrm{th}}=0$, (**a**,**b**) γ=0.05κ, (**c**,**d**) $\varepsilon_{b_{1}}=0.05\kappa$.

**Figure 9 micromachines-13-00591-f009:**
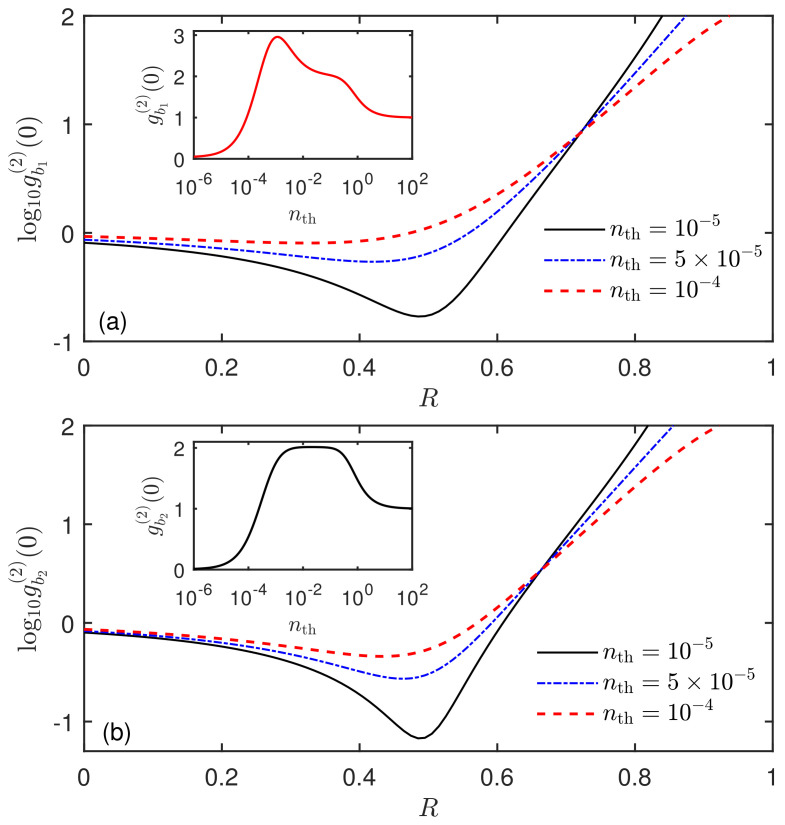
Steady-state equal-time second-order correlation function $g_{b_{i}}^{\left(2\right)}\left(0\right)$ of the SAW modes (**a**) $b_{1}$ and (**b**) $b_{2}$ versus the ratio *R* of two driving strengths with different thermal phonon occupation numbers $n_{\mathrm{th}}$ in the mechanical resonant case Δ=0. The insets of (**a**,**b**) show $g_{b_{1}}^{\left(2\right)}\left(0\right)$ and $g_{b_{2}}^{\left(2\right)}\left(0\right)$ versus the thermal phonon occupation number $n_{\mathrm{th}}$, respectively. The system parameters are the same as in Figure 2 except for g=κ and R=0.5 for the insets.

## Data Availability

Not applicable.

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
