# Peer review of "Enhanced Phonon Antibunching in a Circuit Quantum Acoustodynamical System Containing Two Surface Acoustic Wave Resonators"

_micromachines, 2022, doi:10.3390/mi13040591_

Round 1

Reviewer 1 Report

The authors consider the interesting and timely setup of two surface acoustic wave (SAW) resonators interacting with a single superconducting transmon qubit, within the framework of quantum acoustodynamics (QAD). In particular, they explore the quantum phenomenon of antibunching, in this case, phonon antibunching -as analogue of the well-known quantum optical photon antibunching. They consider the driving of one or of the two resonators and the phenomenon of phonon blockade.

The authors realize a thorough discussion of the literature, putting exhaustively their results into a wider perspective. This effort should be highly appreciated by the interested reader. The paper is clear and well-written. The analysis and methodology is sound- Results seem correct and should be helpful for the interdisciplinary community of quantum technologies.

As a suggestion, I would prompt the authors to consider a discussion of the considered range of parameters in the plots, in terms of state-of-the-art technology --references for the considered values should be enough-- and/or future prospects -in case the values have not yet been achieved.

Overall, I am happy to recommend this paper for publication in Micromachines. 

Reviewer 2 Report

The manuscript presents some theoretical considerations on possible applications of SAW devices in quantum applications. The authors are proposing a ‘schematic theoretical model” on which they are performing quantum physics based calculations on phonon propagation and eventually trying to propose some possible applications. While I do not feel competent in evaluating quantum physics based calculations correctness I do have some comments on the proposed model and proposed possible application.

The main comments are about proposed “experimental setup”

1) While the authors are talking about very precise oscillation modes of the device, they are however introducing IDT’s between the two ‘Bragg mirrors” on both axes and even propose an 'resonator' asymmetric geometry (left-right). From my experience this will determine significant (asymmetric) wave reflection on the propagation surface which should produce a lot of noise and oscillation band enlargement. If we further consider the fact that IDT’s need to have some electric contacts attached, the problem becomes even more critic.

2) The authors are talking about necessity of cryogenic cooling to ‘16.5 mK’ temperatures, while talking about ‘4 GHz’ propagation on a (I guess piezoelectric) ‘physical’ surface, which again it will be performed with losses (It will “physically” take place in a 3D system) and this means temperature increase, local gradiental temperature distributions (from the surface actually) and so on.

3) Proposed equation (1) do not seems to consider any perturbation or asymmetry factor nor the fact that propagation will be effectively talking place in a 3D model with dispersive propagation and so on. In other words, no terms for IDT’s and surface introduced noise is foreseen.

Unless the authors will try at least to ‘tangentially’ address such technical problems, I doubt that these theoretical considerations could be considered as a “proposed scheme” to achieve some practical results.

Reviewer 3 Report

The paper “Enhanced phonon antibunching in a circuit quantum acousto dynamical system containing two surface acoustic wave resonators” by Tai-Shuang Yin, Guang-Ri Jin, and Aixi Chen, describes innovative phenomena. The theoretical basics are thoroughly discussed and interpreted. An analysis of the phenomena can be performed by correlation functions. The results were evaluated by many simulations which are presented in several figures.
Thus, quantum mechanical and classical arrangements can be distinguished in this way. Devices for quantum information technology might be designed with these ideas.

Round 2

Reviewer 2 Report

In the revised manuscript, the authors have tried to consider the engineering challenges in building such a device and added comments on these practical aspects. However, in my opinion, such comments should better be included in the introduction and model description parts rather then conclusions, respectively the authors should try to support their model approximations and clarify its limitation before using it, rather then making all the calculation and addressing that issue in the conclusion part. Also, part of the explanations which authors have given to my questions (e.g "we did not study the effect of different dissipation components separately, but considered the overall dissipation.) might be somehow mentioned to the readers as well in my opinion.
